# Alignement Bilingue des Résumés et des Mots-Clés de `theses.fr`

Ziqian Peng[1,2]    Lichao Zhu[3]    Maxime Bouthors[4]    François Yvon[1]

(1) ISIR, CNRS and Sorbonne Université, Paris, France
(2) ALMAnaCH, Inria, France
(3) ALTAE, Université Paris Cité, Paris, France
(4) Systran by ChapVision, Paris, France
{peng,yvon}@isir.upmc.fr, lichao.zhu@u-paris.fr,
mbouthors@chapsvision.com

RÉSUMÉ

Les développements récents des systèmes de traduction automatique (TA), en particulier l'utilisation de grands modèles de langue exploitant des longs contextes, des exemples et des informations terminologiques, ont conduit à améliorer l'utilisabilité des textes cibles générés automatiquement. Pour accompagner l'utilisation de tels systèmes pour des documents scientifiques, nous documentons dans cette contribution nos efforts pour construire des ressources pour la TA de documents académiques à partir des méta-données de theses.fr.

ABSTRACT

**Bilingual alignments for abstracts and keywords from `theses.fr`**

Recent developments in machine translation (MT) systems, such as the use of large language models able to handle long contexts, in-context examples and terminological information, have led to improvements in the usability of automatically generated target texts. To support the use of such systems for scientific documents, in this paper we document our efforts to build resources for the machine translation of scholarly documents based on metadata from the French archive theses.fr.

MOTS-CLÉS : Alignement de Phrases ; Alignement de Mots-Clés ; Corpus Académiques Bilingues.

KEYWORDS: Sentence Alignment ; Keyword Alignment ; Scholarly Bilingual Corpus.

## 1 Introduction

Les systèmes de traduction automatique (TA) neuronaux de l'état de l'art dérivent de grands modèles de langue multilingues généralistes ; disponibles dans des applications « grand public », ils fournissent des traductions utiles pour une gamme croissante d'applications et de contextes.

La traduction de longs documents reste toutefois un sujet délicat : la modélisation d'un certain nombre de phénomènes discursifs (résolution d'anaphores, modélisation de la cohésion) impose en effet de considérer des contextes de traduction élargis, qui s'étendent au-delà de phrases isolées (Maruf *et al.*, 2021). Pourtant, même si les grands modèles de langues actuels sont capables de prendre en compte de tels contextes étendus, intégrant des centaines, voire des milliers de mots, la traduction de ces grands empans de textes ne donne pas toujours lieu à des améliorations des métriques automatiques de traduction (Wang *et al.*, 2024 ; Wu *et al.*, 2025). Les travaux de recherche les plus récents considèrent

donc des blocs de texte contenant plusieurs phrases, exemplairement un paragraphe, qui semblent fournir le meilleur compromis actuel (Kocmi *et al.*, 2025). Le développement et l'évaluation de tels systèmes demande en complément un effort de constitution de corpus de documents parallèles, nécessaires aussi bien pour l'affinage et le test de systèmes de TA (O'Brien *et al.*, 2025). Des documents parallèles de qualité contrôlée sont également utiles pour produire des traductions à base d'exemples. Disposer d'un ensemble le plus divers possible permet de sélectionner à l'inférence le ou les textes les plus pertinents à insérer dans l'instruction fournie au modèle (Wu *et al.*, 2024).

Pour ce qui concerne la traduction de textes scientifiques, de telles données sont encore relativement rares : les principales initiatives résultent des campagnes d'évaluation de la traduction en domaine biomédical organisées dans le cadre de la conférence WMT depuis 2016 (Bojar *et al.*, 2016), qui ont permis de développer des corpus alignés de résumés pour une dizaine de paires de langues [1] ; pour la paire français-anglais, des résumés de revues systématiques de la littérature sont préparés par (Ive *et al.*, 2016) [2]. AcaData (Lacunza *et al.*, 2026), issu de la fouille archives d'institutions scientifiques, a permis d'augmenter les domaines et sources de données disponibles pour une douzaine de paires de langues européennes, en particulier impliquant l'anglais et l'espagnol, ainsi que le français et l'allemand. À une moindre échelle, les corpus UFAL (Rosa & Zouhar, 2022) et ASCAT (Sibaee *et al.*, 2026) contiennent respectivement 3500 et 500 résumés de documents scientifiques en anglais alignés avec des équivalents tchèques et arabes.

Notre initiative s'inscrit dans le droit fil de ces travaux, en se focalisant sur la paire anglais-français. Elle amplifie les efforts présentés dans (Peng *et al.*, 2026), qui préparent des corpus parallèles pour deux domaines scientifiques. Dans ce nouveau travail, nous moissonnons l'intégralité de la base de données de thèses theses.fr, et en dérivons deux ressources principales : d'une part, un ensemble d'environ 300 000 résumés bilingues parallèles alignés au niveau des phrases. Au-delà de son intérêt propre (par exemple pour calculer des métriques d'évaluation), cet alignement nous permet également de contrôler la qualité globale des alignements de document. D'autre part, un grand ensemble de couples de mots-clés appariés, pouvant servir de glossaire bilingues pour augmenter la traduction en domaine scientifiques avec des traductions de termes du domaine (Moslem *et al.*, 2023). Ces deux ressources sont distribuées sous une licence permissive, accompagnées de toutes les informations déjà présentes dans theses.fr.

# 2   Méthode

## 2.1   La base de données `theses.fr`

La base de données theses.fr [3] recense les thèses soutenues ou en préparation en France depuis 1985. Son développement a connu plusieurs phases successives, intégrant des données et méta-données de plus en plus complètes. La base de données est rendue publique sous licence Etalab [4] ; une extraction est déposée annuellement sur le site data.gouv.fr [5].

Cette base est accessible via une API XML, qui expose des informations complètes relatives à la

---

1. https://github.com/biomedical-translation-corpora/corpora
2. https://github.com/fyvo/CochraneTranslations
3. https://theses.fr/fr/apropos
4. https://alliance.numerique.gouv.fr/licence-ouverte-open-licence/
5. https://www.data.gouv.fr/fr/datasets/theses-soutenues-en-france-depuis-1985/

thèse (école doctorale et discipline, nom de l'étudiant-e, des encadrant-e-s et des membres du jury, dates de première inscription et de soutenance, etc). Nous nous intéressons principalement à enrichir les informations relatives au contenu de la thèse : titre, résumé, et liste de mots-clés. Ces informations sont disponibles en au moins deux langues : le français et l'anglais, le plus souvent.

La première étape de notre chaîne de traitement automatique consiste à moissonner automatiquement les données de `theses.fr`. Notre moissonnage, réalisé le 22 novembre 2025, a permis de récupérer toutes les thèses marquées comme soutenues jusqu'au 9 décembre 2025. Le nombre d'éléments récupérés au terme de cette étape est de 450 744. Nous traitons ensuite séparément les résumés (qui apparaissent dans les champs `abstracts`) et les listes de mots clés libres.

## 2.2 Alignement automatique des titres et résumés

Le traitement automatique des résumés consiste en quatre étapes principales, qui sont décrites ci-dessous.

### Vérification des résumés

La première étape consiste à vérifier séparément la conformité de chacun des deux résumés. Nous commençons par éliminer toutes les thèses pour lesquels l'un des résumés contient moins de 100 caractères.

Nous réalisons une normalisation (NFC) de l'encodage unicode pour faire en sorte que les lettres accentuées et les caractères spéciaux soient représentés de manière homogène à l'aide de la bibliothèque `unicodedata`[6].

Nous calculons ensuite automatiquement la langue de chacun des résumés en utilisant le détecteur de langue associé à l'outil `fasttext`[7]. Si dans l'immense majorité des cas la langue prédite est bien conforme à celle qui est indiquée dans la base de données, nous observons également de nombreux cas où (a) les deux langues sont inversées (le résumé français est en anglais, et vice-versa) ; (b) il manque un des résumés ; (c) les deux résumés sont dans la même langue ; (d) un des résumés est détecté comme étant dans une langue qui n'est ni le français ni l'anglais (p. ex. espagnol ou portugais). Dans le cas (a), les données sont corrigées ; dans les cas (b)–(d), les thèses correspondantes sont filtrées. Au terme de cette étape, nous conservons 294 646 paires de résumés. L'effet du filtrage est visible sur la figure 1, qui témoigne d'une amélioration visible de conformité des données bilingues depuis le début des années 2000. Il est également possible que la diffusion concomitante des outils de TA ait conduit à faciliter la production de résumés parallèles demandés lors du dépôt de la thèse.

### Alignement des résumés

Chacun des résumés est segmenté en phrases par l'outil Trankit[8] (Nguyen *et al.*, 2021), puis les deux textes sont alignés par une version légèrement modifiée par nos soins de BertAlign (Liu & Zhu, 2022), qui prend en charge de manière robuste les alignements de phrases de type « plusieurs-à-plusieurs ». Comme la majorité des systèmes d'alignement automatique de phrases (Tiedemann, 2011), BertAlign repose sur des techniques de programmation dynamique, en utilisant, pour comparer des segments

---

6. https://docs.python.org/3/library/unicodedata.html
7. https://fasttext.cc/, avec le modèle lid.176.ftz (Joulin *et al.*, 2017a,b). Nous fixons un seuil de 0,5 comme la probabilité minimale pour la détection de langue pour les résumés. Pour les titres, nous distribuons la langue détectée avec sa probabilité, utilisée comme score de confiance.
8. https://github.com/nlp-uoregon/trankit

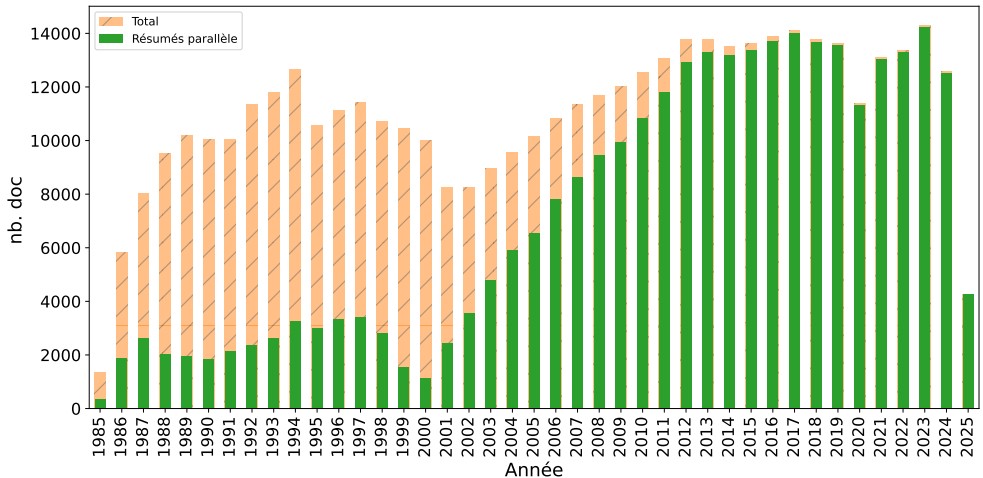

FIGURE 1 – Nombre de résumés (nb. doc) par année, pour tous les résumés collectés (Total) et les résumés parallèles anglais-français. La partie en jaune correspond aux résumés filtrés.

français et anglais, une similarité calculée dans l'espace des plongements de phrases. Ces similarités sont calculées ici par le modèle LaBSE (Feng *et al.*, 2022), une variante du modèle BERT multilingue (Devlin *et al.*, 2019) entraînée pour construire des représentations multilingues pour lesquelles des traductions mutuelles sont proches.

Nous fixons la valeur 0,001 pour le paramètre skip, ce qui améliore les alignements de type « zéro-à-un ». Nous introduisons également un nouveau paramètre, len_slack (avec une valeur de 0,15), qui permet d'éviter d'appliquer une pénalité de longueur aux segments parallèles dont le rapport de longueur est proche de 1 ; cela tend à réduire le nombre d'alignements erronés de type « plusieurs-à-plusieurs ».

**Contrôle de la qualité**

La dernière étape de traitement consiste à vérifier le parallélisme des segments ainsi alignés. Pour cette étape, nous utilisons CometKiwi (Unbabel/wmt22-cometkiwi-da)[9] (Rei *et al.*, 2022), qui est une métrique de traduction automatique dont le calcul ne demande pas de référence, mais qui repose sur des plongements lexicaux entraînés pour reproduire des jugements de qualité humains à partir du couple (phrase source, traduction automatique). Pour chaque couple de phrases, nous calculons son score CometKiwi, puis effectuons une moyenne sur l'ensemble des segments d'un résumé.

À cette étape, nous avons décidé de laisser les utilisateurs du corpus libres de déterminer les résumés qu'ils souhaitent conserver pour des traitements ultérieurs sur la base du score CometKiwi. La valeur seuil de 0,7 est recommandée pour conserver des données de bonne qualité. Un histogramme de ces scores est représenté Figure 2.

**Alignement des titres**

---

9. Avec l'implantation de : https://github.com/Unbabel/COMET.

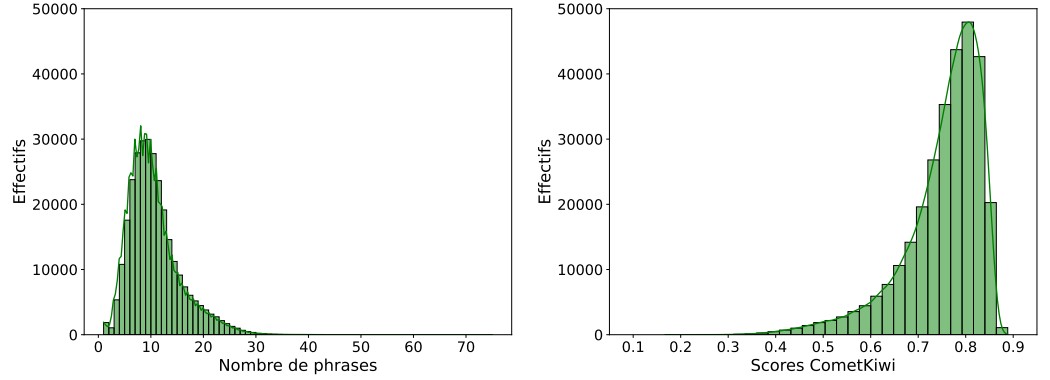

FIGURE 2 – Distribution des nombres de phrases par résumé (à gauche) et des scores CometKiwi (à droite).

La construction d'un corpus de titres parallèles reproduit les mêmes étapes de traitement et de filtrage, à ceci près que l'alignement est obtenu de manière triviale. Après les différentes étapes de filtrage, le nombre de couples de titres conservés pour la paire anglais-français est de 333 227 [10]. Comme pour les résumés, nous laissons les utilisateurs libres de fixer un seuil de qualité (CometKiwi) pour sélectionner les titres parallèles. La distribution des scores associés, en tout point similaire à celle des résumés, est donnée dans la Figure 4.

## 2.3   Extraction de paires de mots-clés

Pour ce travail, nous exploitons les listes de mots-clés *libres* [11], qui sont disponibles à la fois en français et en anglais.

Pour le traitement formel des données, nous avons normalisé l'encodage en Unicode (NFKC), les guillemets, les espaces multiples ainsi que les espaces et la ponctuation en début et en fin de chaîne. Nous avons également procédé à un traitement spécifique pour établir respectivement la forme canonique des mots-clés EN et FR, en retenant la variante la plus fréquente dans l'ensemble des mots-clés. Les traitements automatiques sont les suivants :
— normalisation (NFC) de l'encodage unicode des mots-clés (comme pour les résumés) ;
— identification automatique de langue à partir de la liste des mots-clés monolingues concaténés par des espaces et convertis en minuscules, pour vérifier la langue des mot-clés. Nous fixons un seuil de 0,2 comme probabilité minimale, et nous ne gardons que les paires de mots-clés détectées comme anglais–français ;
— appariement deux à deux des mots-clés à l'aide de l'algorithme de Jonker–Volgenant [12] (Crouse, 2016), sur une matrice de similarité cosinus, calculée à partir des plongements lexicaux LaBSE des mots-clés en langues source et cible, préalablement convertis en minuscules.

---

10. Les deux causes majoritaires de filtrage sont l'absence du titre en anglais (environ 70k thèses), et l'égalité des titres anglais et français (environ 40k thèses).

11. Par opposition aux mots-clés *contrôlés*, qui n'existent qu'en français. Les mots-clés contrôlés sont les entités de type subject auxquels un dcterm, pointant vers un référentiel terminologique, est associé.

12. Nous réalisons cet appariement avec l'implémentation de scipy.optimize.linear_sum_assignment.

Les appariements recevant un score de similarité inférieur à 0,5 sont ensuite filtrés ;
— normalisation des guillemets, des espaces répétés, ainsi que les espaces et ponctuations en début et en fin de chaîne. Nous avons également procédé établi les formes canoniques de mots-clés, en retenant la variante graphique la plus fréquente.
— suppression des doublons.

Nous distribuons, pour chaque paire de mots-clés alignés, les scores de confiance issus de l'identification de langue, la similarité cosinus et l'année de soutenance de la thèse, afin de laisser la possibilité de filtrages ultérieurs.

# 3 Ressources Distribuées

## 3.1 Résumés parallèles

**Statistiques**

Les statistiques principales concernant les résumés alignés sont données dans le tableau 1. Au total, cette ressource contient plus de trois millions de segments parallèles, correspondant à près de 300 000 résumés de thèses.

|         | nb. doc | nb. segments | avg. mots (FR) | avg. mots (EN) |
|---------|---------|--------------|----------------|----------------|
| Titres  | -       | 333 227      | 16             | 14             |
| Résumés | 294 646 | 2 908 119    | 270            | 245            |

TABLE 1 – Nombre de résumés (nb. doc), nombre de segments (nb. segments) et longueur moyenne (avg. mots, délimités par des espaces), pour les résumés et les titres alignés en français (FR) et en anglais (EN).

La figure 3 donne la répartition des thèses par discipline dans notre corpus. On notera que l'information de discipline est imparfaitement normalisée, puisque ce champ contient plus 20 032 références différentes dans theses.fr, de la plus fréquente ("Informatique", 13 102 thèses) aux plus rares ("Sciences du langage, linguistique, français langue étrangère", une thèse).

**Distribution**

Les alignements ainsi calculés et validés sont enfin ajoutés aux autres méta-données disponibles dans theses.fr, ce qui permet de leur associer des informations importantes (discipline, date de soutenance, etc). En pratique, chaque résumé (en français, en anglais) est représenté comme une liste de phrases ; pour chaque couple de phrases, nous fournissons également le ratio de longueur, le cosinus (calculé par LabSE) entre les représentations denses des phrases, et le score CometKiwi. Un exemple de résumé aligné est présenté en annexe dans le Tableau 3. Ces données sont librement téléchargeables sur le site du projet sous la forme d'un fichier parquet [13].

---

13. https://anr-matos.github.io/pages/resources.html

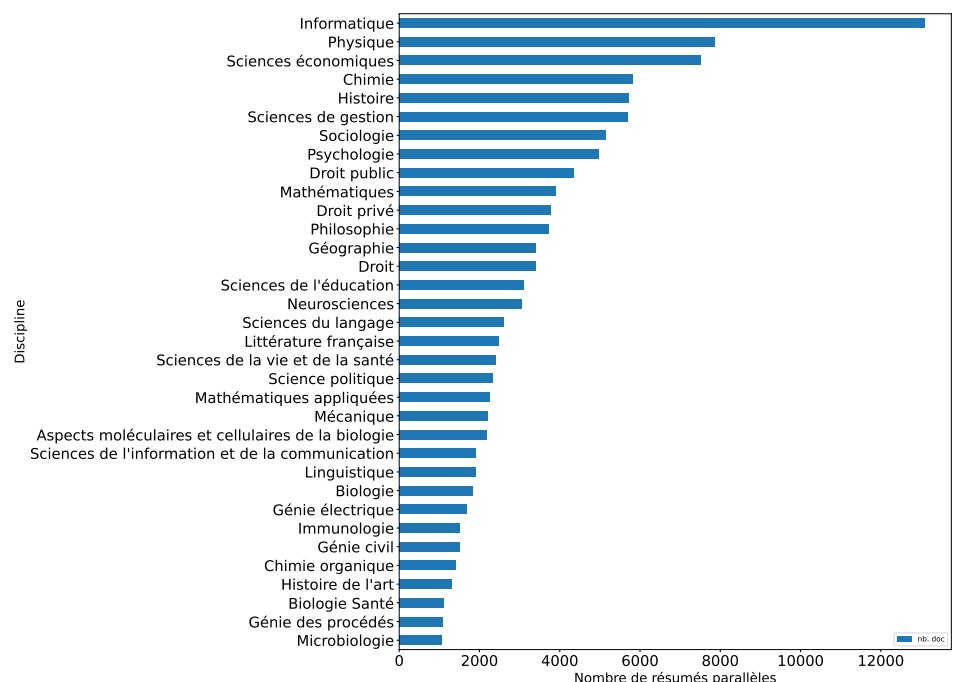

FIGURE 3 – Statistiques des 34 disciplines pour lesquelles on dispose de plus de 1 000 résumés parallèles. Ceci correspond à environ 40% des thèses de notre corpus.

## 3.2 Paires de mots-clés

Le résultat de l'analyse des mots-clés est distribué sous la forme de trois fichiers `.json`. Le premier contient une liste bilingue de 429 833 paires de mots-clés uniques, avec pour entrée la liste des documents, des années, et des scores de similarité cosinus associés à chaque couple de mots. La longueur de ces listes correspond à la fréquence absolue du couple de termes. Ainsi, pour le couple "Scanning Tunneling Microscopy (STM) ; Microscopie à effet tunnel (STM)", on disposera des informations suivantes :

— documents : `[2025MULH7175, 2022AIXM0365]`
— années : `[2025, 2022]`
— cosinus : `[0.8905, 0.8905]`

Ces mêmes données sont également présentées dans deux vues monolingues : l'une qui donne pour chacun des mots-clés anglais la liste complète des équivalents français, avec les mêmes informations (liste de documents, années, scores d'alignement) ; l'autre, symétrique, pour les mots-clés français. L'exploitation de ces deux listes, chacune langue d'environ 340k entrées ( 348 395 en anglais, 342 644 en français) permet d'évaluer la variabilité des équivalences bilingues, et, dans une certaine mesure, de leur évolution temporelle.

Pour illustrer la variabilité des appariements de mots-clés, nous observons que « *Computational linguistics* » est associé à une liste de 13 valeurs, composée de 6 occurrences de « Linguistique computationnelle », de 5 occurrences de « Linguistique informatique », et de 2 occurrences de

« Traitement automatique des langues ».

« *Natural Language Processing* » est bien plus fréquent puisqu'il est utilisé 98 fois, majoritairement associé en français à « Traitement automatique des langues », mais également à « Traitement automatique du language naturel », illustrant de nouveau la variation associée à la dénomination du domaine.

Dans l'alignement des mots-clés, nous découvrons également que la forme de certains mots-clés est problématique du fait de signes de ponctuation (`:`, `;` `|` `/` `()`) qui apparaissent parfois comme dans p. ex. : « *True/false chirality* », « *Raman effect, Surface enhanced* », « *Metal/organic interface* », « *2.5D Carbon/Carbon material (2.5 D C/C)* », etc. Les données EN et FR contiennent respectivement 13 826 et 13 402 mots-clés contenant des signes de ponctuation, ce qui représente pour chaque langue 4% de la totalité des mots-clés dans les deux langues. En revanche, nous n'avons pas proposé de correction en raison du caractère très hétérogène des occurrences.

# 4   Conclusion

Cet article présente les traitements automatiques réalisés pour construire des ressources (titres, documents et mots-clés alignés) utiles à la traduction automatique de documents à partir de l'archive `theses.fr`. La prochaine étape de ce travail consistera à exploiter ces ressources pour améliorer la traduction de résumés de documents académiques pour la paire français-anglais (Tsolakis *et al.*, 2026).

Notre chaîne de traitement, distribuée avec ces nouvelles ressources, permettra de poursuivre la construction de corpus scientifiques alignés en exploitant les méta-données de l'archive `hal.science` ou encore les résumés multilingues disponibles sur la plateforme ISTEX [14].

Parmi les améliorations et extensions possibles de ces travaux, il reste encore à travailler sur la normalisation des mots-clés, et sur la correction automatique des titres et résumés dont certains sont encore mal accentués. Pour ce qui concerne les méta-données, disposer d'une classification (éventuellement hiérarchisée) des disciplines et sous-disciplines serait également souhaitable.

# Remerciements

Ces travaux sont financés par l'Agence Nationale de la Recherche (ANR) dans le cadre du projet Ma-TOS [15] (convention : ANR-22-CE23-0033). Nous remercions également les relecteurs de ARTS pour leurs commentaires, ainsi que tous les membres du projet MaTOS pour les discussions préparatoires à ces travaux.

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

# A   Analyse des disciplines par année

Le tableau 2 présente les trois disciplines les plus fréquentes par année depuis 1985 jusqu'à 2025,
pour les résumés parallèles dans notre corpus. Nous observons que les thèses en Informatique, en
Physique et en Chimie sont dominantes au cours des dix dernières années.

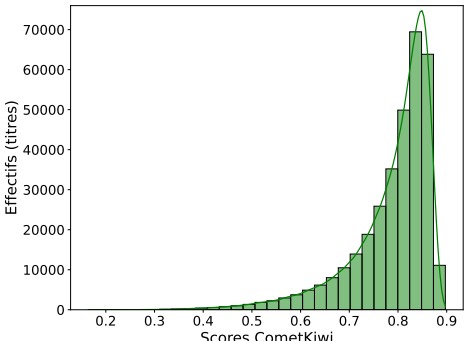

FIGURE 4 – Distribution des scores CometKiwi pour les titres anglais-français.

# B   Analyse des titres parallèles

La figure 4 fait le pendant de la figure 2 et permet de visualiser la distribution des scores CometKiwi
pour les paires de titres extraits de theses.fr. Le mode se situe au delà de 0,85 ; retenir les titres
dont le score excède 0,7 permet de sélectionner des paires de bonne qualité.

# C   Exemple des Données du Corpus

Le tableau 3 donne un exemple d'un résumé aligné de notre corpus, avec les méta-données associées,
par exemple l'année de soutenance de thèse, la discipline et le score CometKiwi. Les données de

chaque paire de résumés sont identifiées avec l'identifiant de thèse correspondant sur theses.fr. De même, les Tableaux 4 et 5 contiennent des exemples des titres et des mots-clés alignés extraits de notre corpus, avec les scores de confiance pour la détection de langue.

| année | top-1 | nb. doc | top-2 | nb. doc | top-3 | nb. doc |
|---|---|---|---|---|---|---|
| 2025 | Informatique | 233 | Physique | 191 | Chimie | 147 |
| 2024 | Informatique | 686 | Physique | 440 | Chimie | 344 |
| 2023 | Informatique | 672 | Physique | 481 | Chimie | 369 |
| 2022 | Informatique | 705 | Physique | 419 | Chimie | 356 |
| 2021 | Informatique | 704 | Physique | 371 | Chimie | 295 |
| 2020 | Informatique | 595 | Physique | 320 | Chimie | 218 |
| 2019 | Informatique | 684 | Physique | 380 | Chimie | 301 |
| 2018 | Informatique | 705 | Physique | 351 | Chimie | 297 |
| 2017 | Informatique | 687 | Physique | 407 | Chimie | 357 |
| 2016 | Informatique | 661 | Physique | 405 | Chimie | 313 |
| 2015 | Informatique | 669 | Physique | 402 | Chimie | 318 |
| 2014 | Informatique | 675 | Physique | 419 | Sciences de gestion | 297 |
| 2013 | Informatique | 718 | Physique | 408 | Sciences de gestion | 284 |
| 2012 | Informatique | 599 | Physique | 466 | Chimie | 299 |
| 2011 | Informatique | 582 | Physique | 297 | Sciences économiques | 259 |
| 2010 | Informatique | 568 | Physique | 258 | Sciences de gestion | 253 |
| 2009 | Informatique | 477 | Physique | 244 | Sciences de gestion | 242 |
| 2008 | Informatique | 511 | Sciences de gestion | 250 | Physique | 246 |
| 2007 | Informatique | 418 | Physique | 219 | Sciences de gestion | 213 |
| 2006 | Informatique | 367 | Physique | 179 | Sciences de gestion | 167 |
| 2005 | Informatique | 283 | Physique | 153 | Sociologie | 136 |
| 2004 | Informatique | 226 | Droit | 132 | Physique | 125 |
| 2003 | Informatique | 144 | Sciences économiques | 118 | Sociologie | 115 |
| 2002 | Physique | 125 | Informatique | 103 | Droit | 98 |
| 2001 | Droit | 111 | Sciences économiques | 92 | Informatique | 73 |
| 2000 | Sciences économiques | 46 | Génie des matériaux | 32 | Psychologie | 32 |
| 1999 | Sciences économiques | 105 | Histoire | 77 | Droit privé | 73 |
| 1998 | Sciences économiques | 218 | Histoire | 211 | Droit public | 150 |
| 1997 | Sciences économiques | 294 | Histoire | 280 | Littérature française | 190 |
| 1996 | Histoire | 250 | Sciences économiques | 242 | Littérature française | 180 |
| 1995 | Histoire | 275 | Sciences économiques | 219 | Littérature française | 180 |
| 1994 | Histoire | 258 | Sciences économiques | 239 | Littérature française | 180 |
| 1993 | Histoire | 258 | Sciences économiques | 187 | Droit public | 113 |
| 1992 | Sciences économiques | 207 | Histoire | 194 | Droit privé | 107 |
| 1991 | Histoire | 165 | Sciences économiques | 144 | Droit public | 110 |
| 1990 | Sciences économiques | 131 | Histoire | 129 | Linguistique | 101 |
| 1989 | Histoire | 153 | Sciences économiques | 137 | Droit public | 110 |
| 1988 | Sciences économiques | 147 | Histoire | 136 | Psychologie | 109 |
| 1987 | Sciences économiques | 168 | Histoire | 161 | Littérature française | 160 |
| 1986 | Sciences économiques | 161 | Littérature française | 100 | Histoire | 99 |
| 1985 | Sciences physiques | 49 | Physique | 23 | Informatique | 19 |

TABLE 2 – Les disciplines de thèse les plus fréquentes (top-1), les secondes (top-2) et le troisèmes (top-3) plus fréquentes par années, pour les résumés parallèles dans notre corpus.

| id | 2023STET0014 |
|---|---|
| year | 2023 |
| discipline | Biologie, Neurosciences et Physiologie végétale |
| fr | ['Le rosier est l'une des plantes ornementales les plus cultivées au monde.' |
| | 'Son parfum est universellement reconnaissable et aimé.' |
| | 'Nous avons donc étudié les composés organiques volatils (COV) qui jouent un rôle dans l'attractivité du parfum des roses.' |
| | 'Dans une première expérience, 10 variétés de roses fraîchement cueillies ont été présentées en aveugle à 20 participants ayant pour consigne de noter les odeurs selon différents paramètres perceptifs.' |
| | 'Nous avons également évalué le comportement moteur des participants reflétant l'attractivité des odeurs de manière inconsciente.' |
| | 'Les COV émis par les roses ont ensuite été capturés par headspace et analysés par GC-MS.' |
| | 'Nous avons ensuite étudié les relations entre les espaces biochimiques, perceptifs et comportementaux et montré en particulier l'importance des oxylipines dans le caractère plaisant des roses.' |
| | 'Ces données originales mettent en lumière la complexité des mécanismes mis en jeu dans la perception d'un mélange complexe d'odorants et l'importance de composés à odeur verte dont le rôle a été jusqu'ici négligé dans l'attractivité du parfum de la rose.' |
| | 'Dans une seconde expérience, nous nous sommes demandé dans quelle mesure la sélection a pris en compte le caractère olfactif et son influence sur la diversité des allèles des gènes impliqués dans les voies de biosynthèse des COV.' |
| | 'Nous avons étudié la diversité allélique des gènes NUDX1-1 et PAAS dans 60 cultivars de roses.' |
| | 'Cette diversité des allèles s'est révélée faible ce qui pourrait suggérer une faible diversité génétique des croisements réalisés par les obtenteurs ou pourrait également être dû à la prise en compte du caractère « parfum » dans le processus de sélection.' |
| | "] |
| en | ['The rose is one of the most cultivated ornamental plants in the world and has been since antiquity.' |
| | 'In addition to its colors and shapes, which are particularly appreciated and valued in painting and poetry, its fragrance is universally recognizable and loved.' |
| | 'During this thesis, we studied the volatile odorant compounds that play a role in the attractiveness of the rose fragrance.' |
| | 'Thus, in a first experiment, ten varieties of freshly picked roses were presented blindly to twenty participants who were instructed to rate the odors according to different perceptual parameters (hedonicity, familiarity, etc.).' |
| | 'In parallel to these assessments, we evaluated, by video-tracking, the motor behavior of the participants reflecting the attractiveness of the odors in an unconscious manner.' |
| | 'The volatile odorant compounds emitted by the different roses were then captured by headspace and analyzed by gas chromatography-mass spectrometry.' |
| | 'In this context, we studied the relationships between biochemical, perceptual, and behavioral spaces and showed the importance of oxylipins in the pleasantness of roses.' |
| | 'These original data highlight the complexity of the mechanisms involved in the perception of a complex mixture of odorants and the importance of green-smelling compounds whose role in the attractiveness of rose scent has been neglected until now.' |
| | 'In a second experiment, we asked to what extent selection has considered the olfactory character and whether this has influenced the allele diversity of genes involved in the biosynthetic pathways of olfactory compounds.' |
| | 'We therefore studied the allelic diversity of the NUDX1-1 and PAAS genes in sixty rose cultivars of different geographical origins.' |
| | 'This allelic diversity was very low, which could suggest a small genetic diversity of crosses made by breeders. This result could also be due to the consideration of the fragrance character in the selection process.' |
| | 'This work was financed by the Ambition Research Pack of the AURA Region and supported by the CNRS, INSERM and the University Jean Monnet of Saint-Etienne.'] |
| has_empty_align | True |
| nb_sent | 12 |
| d-cometkiwi | 0,72 |
| cometkiwi22 | [0,75 ; 0,51 ; 0,75 ; 0,77 ; 0,75 ; 0,74 ; 0,82 ; 0,84 ; 0,82 ; 0,73 ; 0,82 ; 0,37] |
| bertalign | [0,46 ; 0,21 ; 0,39 ; 0,41 ; 0,36 ; 0,36 ; 0,40 ; 0,39 ; 0,38 ; 0,43 ; 0,42 ; -0,00] |
| length_ratio | [0,75 ; 0,34 ; 1,04 ; 0,93 ; 0,78 ; 0,63 ; 1,17 ; 1,13 ; 1,08 ; 0,76 ; 1,28 ; -100,00] |
| cos | [0,79 ; 0,62 ; 0,71 ; 0,80 ; 0,70 ; 0,71 ; 0,84 ; 0,89 ; 0,89 ; 0,86 ; 0,82 ; 0,25] |

TABLE 3 – Une paire de résumés en français (fr) et anglais (en) segmentés et alignés de notre corpus, avec les méta-données associées, y compris l'année de soutenance (year), la discipline, l'existence d'un l'alignement zéro-à-un (has_empty_align), le nombre des phrases (nb_sent), les scores CometKiwi de chaque phrase et leur moyenne (d-cometkiwi), les scores d'alignement produits par bertalign, le ratio de longueur source / cible par caractère (length_ratio) qui est -100 par convention pour l'alignement zéro-à-un, et les similarités cosinus (cos). Les scores ont été arrondis à deux décimales pour faciliter la lecture.

| id | fr | en | lid_prob_fr | lid_prob_en | cometkiwi22 |
|---|---|---|---|---|---|
| 2020COAZ6034 | Influences biochimiques, anatomiques et cognitives de la troncation exopeptidasique N-terminale du peptide A$\beta$ | Biochemical, anatomical, and cognitive influences of exopeptidases truncation in N-terminal A$\beta$ peptide | 0.7119 | 0.8314 | 0.8249 |
| 1987PA030145 | L'expression de l'oralite dans l'ecriture de la negritude | Expression of orality in the writing of negritude | 0.8381 | 0.7358 | 0.6136 |
| 2004PA030100 | Astérix : une bédélecture en français langue étrangère pour italophones | Asterix : a reading cartoon in French as a foreign language for Italian speakers | 0.9917 | 0.9003 | 0.7764 |
| 2014MON1T010 | Régulations monoaminergiques AMPc-dépendantes du coeur sain et pathologique | cAMP-dependent monoaminergic regulations of the healthy and failing heart | 0.9205 | 0.7833 | 0.7857 |
| 2015MONTS246 | Apport du GPS pour la quantification des déformations extrêmement lentes et mouvements verticaux dans les chaînes de montagnes françaises | Contribution of GPS for the quantification of extremely slow deformations and vertical movements in the French mountains chains | 0.9902 | 0.8958 | 0.8405 |

TABLE 4 – Des exemples des titres français (fr) et anglais (en) alignés tirés de notre corpus, avec l'identifiant unique la de thèse dans `theses.fr` (id), les scores de confiance de la détection automatique de langue (lid_prob) et les scores CometKiwi.

| id | en | fr | cos-uncased | year |
|---|---|---|---|---|
| 2022CYUN1098 | Population dynamics | Dynamique de populations | 0.9229 | 2022 |
| 2025SORUL011 | Baudelaire | Baudelaire | 1.0000 | 2025 |
| 2020UPASS115 | Model Based Testing | Model Based Testing | 1.0000 | 2020 |
| 2012GRENS008 | Memory Self-Efficacy | Sentiment d'Auto-efficacité Mnésique | 0.6974 | 2012 |
| 2012LYO10288 | Computer vision | Vision par ordinateur | 0.9336 | 2012 |
| 2011STET4036 | Poly(ethylene glycol) | Poly(éthylène glycol) | 0.9721 | 2011 |
| 2020AIXM0609 | Agent-Based modelling and simulation | Modélisation et simulation basées sur des agents | 0.9439 | 2020 |
| 2023UPSLC013 | Headspace extraction | Extraction par espace de tête | 0.9163 | 2023 |
| 2024UMONG104 | Population genetics | Génétique des populations | 0.9517 | 2024 |
| 2018LYSEI036 | Properties | Propriété | 0.8348 | 2018 |
| 2023UBFCI010 | Neurodegeneration | Neurodégénérescence | 0.9451 | 2023 |
| 2019STRAE002 | Perovskites | Pérovskites | 0.7362 | 2019 |
| 2015LYO22009 | South America | Amérique du Sud | 0.9000 | 2015 |
| 2019STRAJ104 | Hippocampus | Hippocampe | 0.8532 | 2019 |

| | |
|---|---|
| id | 2020UPASS115 |
| en | ['Model Based Testing', 'Model Transformation', 'Model Driven Engineering', 'Sequence Diagram'] |
| fr | ['Model Based Testing', 'Transformation de modèles', 'Ingénierie dirigée par les modèles', 'Diagramme de séquence'] |
| cos-uncased | [1. ; 0,93846929 ; 0,88673204 ; 0,92561126] |
| keywords_fr | Model Based Testing///Transformation de modèles///Ingénierie dirigée par les modèles///Diagramme de séquence |
| keywords_en | Model Based Testing///Model Driven Engineering///Model Transformation///Sequence Diagram |
| lid_keywords_fr | fr |
| lid_keywords_en | en |
| prob_lid_keywords_fr | 0.934740 |
| prob_lid_keywords_en | 0.955011 |
| year | 2020 |

TABLE 5 – En haut : des exemples de mots-clés anglais (en) et français (fr) alignés extraits de notre corpus, avec l'identifiant de thèse correspondant (id), la similarité cosinus LaBSE calculée en minuscule (cos-uncased), et l'année de soutenance (year). En bas : un exemple des méta-données complémentaires dans notre corpus, avec les résultats de détection de langue.