# OpenReview forum: "Alignement Bilingue des Résumés et des Mots-Clés de these.fr"
_ls2n.fr/CORIA-TALN/2026/Workshop/ARTS — ls2n CORIATALN 2026 Workshop ARTS Submission_

### Official Review · Reviewer_uWWV · 2026-04-30

**Mode De Presentation:** Oral

**Confience:**

Oui

**Decision:**

Accepté

**Relecture:**

La méthode d’alignement permettant d’aboutir à un corpus de résumés et de titres parallèles alignés est décrite de manière détaillée. En revanche, les traitements réalisés sur les paires de mots-clés sont listés et peu développés. Quelques détails sur l’évaluation de la similarité et sur les problématiques associées seraient bienvenus.

Quelques coquilles relevées :
* 1 : la traduction en domaine scientifiques => domaine**s**
* 2.2 : toutes les thèses pour lesquels => lesquel**le**s
* 2.3 :
  * note 10 : a fait émerger certains termes problématique => problématique**s**
  * en langues source et en cible => en langue source et en langue cible OU en langues source et cible
* 3.2 :
  * ;Microscopie à effet tunnel => manque un espace après le point-virgule
  * ,364 638 en français) => manque un espace après la virgule
  * en deça => en deç**à**
* 4 :
  * Note 12 : URL site Istex => un « https:// » en trop
* Annexe A / Table 2 : et le troisèmes (top-2) plus fréquentes par années => et le**s** trois**i**èmes (top-**3**) plus fréquentes par anné**e**
* Annexe B : au delà => au-delà (avec trait d'union)

**Resume:**

Cet article porte sur la construction de ressources alignées à partir des métadonnées de *theses.fr* pour alimenter des systèmes de TA de documents académiques.
Il décrit la méthode d’alignement permettant d’aboutir à un corpus de résumés et de titres parallèles alignés et présente les traitements réalisés sur les paires de mots-clés extraites des métadonnées.
L’article décrit ensuite le contenu de ces ressources telles qu’elles sont distribuées sur le site du projet et ouvre sur une perspective d’enrichissement de ces corpus par d’autres sources (hal.science et Istex).

---

### Official Review · Reviewer_dMdn · 2026-05-05

**Mode De Presentation:** Oral

**Confience:**

Oui

**Decision:**

Accepté

**Relecture:**

L'article est très bien écrit avec beaucoup de précisions dans la description des traitements informatiques. La ressource produite qui sera distribuée, sera très utile pour la communauté qui s'intéresse au traitement des documents scientifiques, notamment en traduction automatique. Le point d''amélioration que je vois serait d'avoir une évaluation de la qualité de la ressource un peu plus poussée. Les auteurs proposent pour chaque paire alignée des métriques automatiques de qualité et les utilisateurs de la ressource sont laissés libres ensuite de la sélection. Il pourrait être intéressant d'évaluer à quel point les métriques produites sont fiables par exemple.

**Resume:**

L'article présente une méthode pour construire une ressource bilingue à partir des documents académiques de theses.fr en exploitant les meta-données produites à la fois en anglais et en français (titre, résumé, mots-clés). Les traitements automatiques proposés permettent d'obtenir des titres alignés fr-en, des résumés alignés par phrase en mode "plusieurs-à-plusieurs", des mots-clés alignés. Cette ressource sera ensuite exploitée pour améliorer la traduction automatique (fr-en) des documents académiques.

---

### Decision · Program_Chairs · 2026-05-07

Accept (Oral + Poster)